# When Personalization Is Not an Option: An In-The-Wild Study on Persuasive News Recommendation

**Cristina Gena [1], Pierluigi Grillo [1], Antonio Lieto [1,2], Claudio Mattutino [1] and Fabiana Vernero [1,***

[1]   Dipartimento di Informatica, Università degli Studi di Torino, 10149 Turin, Italy; cristina.gena@unito.it (C.G.); pierluigi.grillo@unito.it (P.G.); antonio.lieto@unito.it (A.L.); claudio.mattutino@unito.it (C.M.)

[2]   ICAR-CNR Institute, 90146 Palermo, Italy

*   Correspondence: fabiana.vernero@unito.it; Tel.: +39-011-670-6758

**Abstract:** Aiming at granting wide access to their contents, online information providers often choose not to have registered users, and therefore must give up personalization. In this paper, we focus on the case of non-personalized news recommender systems, and explore persuasive techniques that can, nonetheless, be used to enhance recommendation presentation, with the aim of capturing the user's interest on suggested items leveraging the way news is perceived. We present the results of two evaluations "in the wild", carried out in the context of a real online magazine and based on data from 16,134 and 20,933 user sessions, respectively, where we empirically assessed the effectiveness of persuasion strategies which exploit logical fallacies and other techniques. Logical fallacies are inferential schemes known since antiquity that, even if formally invalid, appear as plausible and are therefore psychologically persuasive. In particular, our evaluations allowed us to compare three persuasive scenarios based on the *Argumentum Ad Populum* fallacy, on a modified version of the *Argumentum ad Populum* fallacy (*Group-Ad Populum*), and on no fallacy (neutral condition), respectively. Moreover, we studied the effects of the *Accent Fallacy* (in its visual variant), and of positive vs. negative *Framing*.

**Keywords:** persuasive technologies; fallacies; framing; non-personalized recommender systems; evaluation in the wild

---

## 1. Introduction

The study of persuasive interaction between humans and digital artifacts is presently a well-established area of research in Human–Computer Interaction (HCI) [1]. Persuasive technologies can adopt several strategies to change the attitudes and behaviors of their users, depending on the main role they play—whether as *tools* that make activities easier or more efficient to perform, as *media* that provide virtual environments and allow users to rehearse some target behaviors, or as *social actors* that elicit emotional responses, similarly to human–human interaction [1]. Personalization technologies such as recommender systems can be considered persuasive in that they help users to efficiently retrieve potentially relevant content, thus increasing the likelihood that users will eventually perform some target action, for example buying a product or reading a piece of news. In fact, it has long been known that personalized content raises more attention [2], an important precondition for effective persuasion, and some authors even claimed that the mere belief that some content is personalized pushes users to consider it as more relevant [3]. Following these ideas, several studies have shown that personalized persuasive technologies are more effective than one-size-fits-all approaches when it comes to achieving a certain goal [4], in contexts as diverse as education [5], workplace habits [6], and health [7,8].

Some recommender systems, however, may be designed (e.g., for practical or juridical reasons) to provide their users with non-personalized recommendations [9,10]. As stated by Ricci et al. [11], for example, non-personalized recommendations are typical of magazines and newspapers. In fact, they can be useful for online information providers that aim at granting the widest possible access to their content, and therefore choose not to have registered users, but at the same time want to recommend items to their users. A possible solution can be represented by product association rules, as will be described in Section 2.3, but there can be also other needs. For instance, non-personalized recommenders might suggest popular, newly inserted or featured items, they might advertise the choices of some editorial board or try to promote niche contents users are likely to be unaware of.

In this paper, we focus on the case of non-personalized recommender systems, and explore persuasive techniques that can be used to enhance recommendation presentation and thus capture the user's interest on suggested items leveraging the way news is presented and then perceived, with the implicit aim of increasing the number of user clicks on them. To this end, we start from the hypothesis, first formulated by Lieto and Vernero [12], that there is a strong connection between logical fallacies (forms of reasoning which are logically invalid but cognitively effective, studied since the antiquity in the fields of logic and rhetoric) and some of the most common persuasion strategies adopted within digital technologies. We present the results of two large-scale "evaluations in the wild", carried out with real users on a real online magazine, UnitoNews (https://www.unitonews.it) and based on data from 16,134 and 20,933 user sessions, respectively. To the best of our knowledge this is the first extensive evaluation on real data carried out in the field of non-personalized persuasive recommendations based on the exploitation of both logical fallacies and framing techniques.

In our first evaluation, we have annotated recommendations with three different persuasive sentences (recommendation explanations), two of them based on fallacies that exploit the persuasive potential of popularity, i.e., the *Argumentum Ad Populum* fallacy and a modified version of the *Argumentum Ad Populum* fallacy (that we called *Group-Ad Populum*), and one based on no fallacy at all (neutral condition). Moreover, we studied the effect of the *Accent fallacy* (in its visual variant) in combination with all the above-mentioned fallacies.

In our second evaluation, we evaluated the influence of the *Framing effect* [13] in news recommendation, again annotating recommendations with two different sentences, one highlighting a prospective "win", at least in terms of information access (*positive framing*), and one highlighting a prospective "loss" (*negative framing*).

Our results show that both the standard *Argumentum Ad Populum* and the *Group-Ad Populum* fallacies, despite their wide use, do not cause any improvement with respect to the neutral condition, thus confirming previous results on the inefficacy of recommendations based on the mere "appeal to the majority" [14,15], but in contrast to other relevant literature [16–19]). Finally, we found that *negative framing*, combined with visual accent, is effective in improving the number of clicks on recommended news, consistently with our expectations. All the above-mentioned techniques are explained in detail in Section 3.

## 2. Related Work

The research areas related to the work described here include: social navigation (see Section 2.1), recommender systems as persuasive technologies (see Section 2.2) and non-personalized recommender systems (see Section 2.3). We provide in the following an overview of the related work in all these areas by outlining its connection with our work.

### 2.1. Social Navigation

Social navigation is a phenomenon that has its roots in real life, and can be defined as a natural tendency of people to follow each other, by making use of direct and indirect cues about the activities of others [20]. The concept of social navigation has been introduced by Dourish and Chalmers [21], and was defined as the navigation towards a cluster of people or navigation because other people have

looked at something. *Social navigation* can be seen as opposed to *general navigation* and their difference can be compared, for instance, to the difference existing between reading a sign at the airport to find the baggage claim or talking to a person at the airport help desk to find the baggage claim [22].

The effects of social influence have their roots in social psychology. In his past studies, Cialdini [23] identifies the principle of social witness, i.e., the process in which individuals observe others in order to decide what behavior to adopt, and this is one of the most effective methods to persuade people to obtain a specific behavior. This principle has the same meaning also in an online environment that, because of its unfamiliar and uncertain nature, may benefit of navigation aid tools.

Early examples of social navigation can be found in the work of Hill and his colleagues. Hill et al. [24] started from the idea that information on the use of digital resources can guide work, and described two document processing applications (*Edit Wear* and *Read Wear*) where the documents' screen representation was modified to portray their authorship or readership history, by analogy with the physical world, where use leaves wear. Tapestry, the first recommender system, allowed users to express their liking for items, and generated personalized suggestions based on similarities among such evaluations (*collaborative filtering* approach) [25], thus automating the reasoning behind social navigation.

In our evaluation on the persuasive effect of fallacious structures (see Section 5.1), our use of annotations based on the *Group-Ad Populum* and *Argumentum Ad Populum* fallacies can also be considered a form of social navigation.

## 2.2. Recommender Systems as Persuasive Technologies

Persuasive techniques have been widely studied in connection with recommender systems.

As pointed out in Section 1, personalized recommender systems in general can be considered examples of persuasive technologies, making use of the tailoring technique [1] to influence users' behavior. According to this perspective, the persuasive power of personalization is exploited in the selection of the items that are presented to target users (end). However, some authors have recently pointed out that personalization might also be used to improve the way suggested items are presented, and, in particular, the choice of persuasion strategies themselves (means).

For example, Berkovsky et al. [26] suggest that information relating to the target user could be exploited to personalize assistive features (e.g., by delivering information about aspects which are relevant to a certain user), messages (e.g., by tailoring their content and/or presentation) and strategies (e.g., by selecting the persuasive techniques and methods that are more suitable for a certain individual). In a similar vein, Kaptein et al. [4] propose that persuasive technologies can vary the influence principles they exploit depending on their effectiveness for each individual user. According to the authors, information on effectiveness can be obtained either explicitly, e.g., using questionnaires such as the susceptibility to persuasive strategies scale (STPS) [27], or implicitly, e.g., by analyzing user responses to persuasion attempts, and be stored in "persuasion profiles".

According to Tintarev and Masthoff [28], persuasion is one of the goals of recommendation explanations, i.e., the sentences that are used to annotate recommendations in an attempt to communicate the reasoning process behind recommendation generation and therefore to make the system more transparent [29]. Gkika and Lekakos [16] compared the persuasive effectiveness of explanations based on Cialdini's Influence Principles [30], i.e., reciprocity, scarcity, authority, social proof, liking and commitment, and found that all of them were able to produce a shift in users' behavior, namely in their intention to watch a recommended movie. Zanker and Schoberegger [31] also compared the persuasive effectiveness of three different types of explanations providing the same content, but with different styles: "facts" (e.g., "low altitude, easy distance, very family-friendly"), "logical arguments" (e.g., "low altitude, easy distance *therefore* very family-friendly"), and "sentences" (e.g., "This route is of low altitude and easy distance, therefore it is very family-friendly."). The authors found that fact-based and argument-based explanations had a stronger effect on user preferences than full

sentences, possibly because processing sentences is more cumbersome, and the persuasive impact of the "therefore" keyword, which characterizes argument-based explanations, is moderated.

More broadly, Yoo et al. [32] note that a proper and efficient use of recommender systems is only possible if users have confidence in the suggestions they offer. They state that interactions with recommender systems can be best described as conversations that should be analyzed from a communication point of view, and therefore treat recommenders as "sources with the need to persuade their users". From this point of view—and making reference to Fogg's Functional Triad [1]-recommender systems mainly play the role of social actors which can elicit social responses and exploit their human-like qualities to persuade their users. For instance, Yoo and Gretzel [33] start from Fogg [1]'s consideration that source credibility is relevant whenever computers provide advice, and experimentally assess their hypothesis that recommenders perceived as more expert and trustworthy will be preferred. Based on their results, they suggest enhancing recommender systems with cues that can boost credibility perception.

In our evaluations, the persuasive sentences we used to annotate recommendations can be intended as explanations, therefore our work can be closely related to the above-mentioned studies on explanation persuasiveness. In our explanations, we specifically focused on fallacious arguments (see Section 5.1) or sentences using differently polarized frames (see Section 5.2).

### 2.3. Non-Personalized Recommenders

Most recommender systems exploit well-known techniques such as content-based and collaborative filtering in order to provide users with personalized suggestions [34], based on the ideas that they will like similar items to those they liked in the past (content-based filtering), or items that are appreciated by other users with similar tastes (collaborative filtering). Non-personalized recommenders [9,10,35], on the contrary, offer the same recommendations to all their users. For example, they select items to recommend based on criteria such as freshness, some editor's choice, item features (e.g., brand, author, product category), product associations or, more often than not, popularity.

Many commercial recommender systems and marketplaces such as Amazon (https://www.amazon.com/), TripAdvisor (https://www.tripadvisor.com/), IMDB (http://www.imdb.com/), or SoundCloud (https://soundcloud.com/) suggest popular items or annotate them with visual cues representing the average users' ratings.

In the literature, different studies support the idea that highlighting item popularity can influence users' behavior. For example, Salganik et al. [17] carried out an experiment in the music domain and showed that displaying download counts pushed users to prefer popular songs and disregard the least popular ones. Similarly, Zhu and Huberman [18] found that users, who were first asked to choose their favorite options from pairs of items, were likely to change their choices and conform to other people's preferences when they were shown information on item popularity. Su et al. [19] analyzed friend recommendations in Twitter, and found that they were more likely to be accepted if they suggested popular users. Conversely, McNee et al. [36] pointed out that recommending a very popular item can be worthless from the final users' perspective: in fact, most of them will already be aware of it, and they will have either already enjoyed it, or knowledgeably chosen not to do so.

Non-personalized recommenders that make use of product associations [37] suggest items related to what the current user is viewing or buying, based on rules such as "product X and product Y are frequently bought together" [38,39]. Such rules are usually mined from large datasets and capture the fact that users often buy or view a certain set of items during a single browsing session. Making reference to Schafer et al. [35]'s recommender systems taxonomy, these recommendations can be described as *ephemeral*: in fact, they are generated based on information from a single user session, and do not require any data from past interactions on the part of the same user/customer. Thus, there is no need to maintain user models, nor to recognize users, and recommendations are the same for all users displaying the same behavior. As with association rules, item-to-item correlation recommender

systems can also be ephemeral, since they can generate suggestions based on currently selected items, without making reference to historical data about the current user [35].

Following this line of research, Chen et al. [40] show how association rules can be exploited to devise a recommender system for small online retailers that typically have limited processing power to perform complex analyses. Similarly, Mican and Tomai [41] use association rules to model connections between the pages of a web site and suggest relevant content.

In the here presented evaluations, we experimented with non-personalized recommendations based on popularity (see Section 5.1) and freshness (see Section 5.2). In our case study system, we also suggest news based on item–item relationships depending on item features (shared category or descriptive tags), which were not, however, taken into account in our present study.

## 3. Background: Fallacies, Framing and Persuasive Technologies

Logical fallacies are a particular class of non-deductive inferences studied in logic since the antiquity [42,43] (the first classification of such reasoning schema goes back to Aristotle in the *De Sophistichis Elenchis* [44]). They, in fact, enjoy a special status in that, even if invalid from a formal point of view, appear as plausible and therefore are psychologically persuasive [45,46]. Fallacious arguments should not be considered necessarily irrational. Indeed, in the context of ecological approaches to rationality and cognition, it has been pointed out that a fallacious argument is usually an invalid argument endowed with psychological plausibility and a proper heuristic value [47]. During the centuries different research areas such as logic, rhetoric and argumentation theory dealt with the problem of fallacies, pointing out that they are suitable techniques for achieving persuasive goals [43].

B.J. Fogg coined the term *captology* (*Computers As Persuasive Technologies*) in the 1990s, with the aim to describe a new research area which views computer technologies as potential persuaders and concentrates on both their design and their analysis [1].

This area is presently commonly referred to as "persuasive technologies". In this field, the above-mentioned connection between fallacies and technology-based persuasion has been first pointed out by [12,14], where the authors created a persuasion matrix mapping some well-known fallacious arguments to some design features available in websites and mobile apps. This kind of theoretical grounding has also directly inspired the current empirical investigation since the connection with the huge theoretical background provided by the disciplines that, over the centuries, have dealt with all the major aspects of fallacious arguments (in logic, rhetoric, and persuasion) represents a unique and reliable source of knowledge to exploit for the study and design of "computer-driven" persuasion mechanisms.

In the rest of this section, we present the identified connections between some well-known logical fallacies and some of the techniques used in the field of persuasive technologies and relevant for the present work. In this work we have focused our attention only on a subset of the fallacious matrix proposed in [14]. We have considered: *Argumentum Ad Populum* (in two different settings) and accent. We devise to future work the testing of different more complex persuasive schemata based on a mix of different fallacy-based strategies.

The logical fallacy known as "appeal to the majority" (or *Argumentum Ad Populum*), consists of accepting a certain thesis based on the mere fact that the majority of people accept it. A typical characterization of such a fallacy is: "Most people think that X is true/false, then X is true/false" (where "X" can be any statement). This fallacy can be compared to those strategies, commonly used in the realm of persuasive technologies, which owe their persuasive potential to the exploitation of social dynamics. In particular, Fogg refers to well-known social psychology theories (e.g., social comparison and conformity [48]), which can be extended to include computer technologies. According to social comparison theory, people who are uncertain about the way they should behave in a situation proactively collect information about others and use it to build their own attitudes and behaviors. By contrast, conformity theory focuses on normative influence, stating that people who belong to a group usually experience a pressure to conform to the expectations of the other group members.

In the present work we have tested the efficacy of this fallacy in its standard form (explained above) and in a modified version trying to convey the majority effect in specific groups. We call this version the *Group-Ad Populum*.

An additional commonality between fallacies and persuasive technologies regards the debate about the perceived credibility in both digital artifacts and human beings [49]. Such credibility is known to be affected by the so-called halo effect [50], according to which a positive evaluation on a specific aspect (e.g., physical attractiveness) produces a halo which determines an extension of such an evaluation to other, unrelated, aspects (e.g., expertise in a certain field). This is exactly the case of the fallacy relying on the so-called "appeal to the authority" (also *Argumentum Ad Verecundiam*, see [14] for details). It refers to cases of inappropriate transfer where some theses are assumed to hold merely because the people asserting them are wrongly assumed to be authorities about a certain topic due to their achievements and fame obtained in other, unrelated, fields.

Technologies implementing personalization techniques can also be considered persuasive because they provide their users with the information they are most likely to find interesting, based on their personal preferences, aims, skills and experience. Such techniques can be regarded as fallacious because they assume that (i) people will show the same preferences they showed in the past also in the future (content-based filtering) or that (ii) people who showed similar preferences in the past will maintain this similarity also in the future (collaborative filtering), which, although being reasonable, cannot be taken for granted. Personalized information does not only save users the effort to examine a huge amount of content, but it is also more likely to draw their attention and, in case the system suggestions are accepted, it can cause longer-lasting and deeper changes. Tailoring is somehow similar to the so-called audience agreement technique, a well-known strategy in rhetoric and theory of argumentation [14,43] which suggests that persuaders should only use arguments that were already accepted by their audience in order to be effective.

The *Accent fallacy* [44,51], which occurs when a particular emphasis on a part of a sentence is used to manipulate the actual meaning of a proposition, is commonly adopted with a persuasive intent in computer technologies, especially in its visual variant where certain elements are made more visually prominent in order to emphasize (or de-emphasize) them. A common example of the (visual) accent fallacy occurs when special offers (e.g., discounts) are highlighted with big fonts and bright colors, while the possibly restrictive conditions to enjoy them are made scarcely visible. This kind of presentation is fallacious since the inference drawn by the users is than one of considering relevant the emphasized information (e.g., the *suggested* conclusion is: take the special offer!) and not relevant the de-emphasized one (in our example: the restrictive constraint conditions). It is worth-noticing that the mere use of color or of different fonts to highlight a particular aspect of a text or of an interface does not constitute, *per se* a fallacy. We are in the presence of a visual accent fallacy only in the case in which the element put in evidence (or voluntarily hidden) has the goal of driving the users towards a conclusion (e.g., "buy the book X instead of Y") that is not logically justified by the premises.

The accent fallacy can be compared to the concept of misplaced salience in Human–Computer Interaction, which is known as one of the factors limiting situation awareness [52] due to the emphasis it provided to irrelevant cues, leading users to confusion activities and inappropriate behaviors.

Finally, another well-known persuasive technique that is not fallacious *per se* but is based on some well-known cognitive biases in human decision making is the so-called *framing* effect [13,53]. It refers to the role of the context in shaping people's decisions. In fact, using a particular wording instead of another might determine a different configuration of a given problem that consequently, may lead to a given interpretation of a sentence's meaning. A classical example of framing for providing information about food nutrition is the following: "The food X contains 60% of lean meat. Therefore, X is sustainable". Of course, this interpretation is misleading since the conclusion cannot be drawn from the premises, but it is framed by them (e.g., the reverse of the frame, in fact, is that The food X contains 40% of fat meat). Another well-known corollary of the framing effect consists of the fact that there is an asymmetry between prospective losses or wins in the people choice's

architecture [13]. This effect is known as prospect theory and can be roughly explained as follows: people prefer prospective choices that lead them not to lose something instead of choices that could provide them with the possibility of winning something else. This means that framing a given context as a possible loss (negative framing) should be a more sensitive and persuasive move to induce people towards a given behavior. In our analysis we will take for granted that the framing effect and its corollary, e.g., the loss aversion effect, exist as demonstrated by dozens of studies coming from the psychology of reasoning and from the behavioral economics. We will just focus, in this paper, on the evaluation of the persuasive efficacy of these effects when used in a real technological environment. For the purposes of this paper, it is also worth-noticing that the framing effect, based on the loss aversion predicted by the prospect theory, is somehow related to another well-known effect: the scarcity one. In this setting, this means that the more something is perceived as scarce the more the prospective loss is valued as problematic (and this usually leads to a less risk-seeking behavior [54,55] or, in our case, to an action aimed at removing this sense of potential loss based on the scarce availability of news). We refer to Sections 5 and 6 for a discussion of the effect of the negative framing based on this induced sense of scarce availability and potential loss of a given information. Recently the exploitation of this kind of cognitive tendencies, in order to design and evaluate the effect of design elements in guiding people choice in digital environments, is gaining widespread attention in the area of Human–Computer Interaction [56]. Schneider et al. [57] call this area Digital Nudging. In our opinion, our work can be ascribed to this class of analysis. We want to point out, however, that our analysis considers additional decision strategies (e.g., interface schemata based on the logical fallacies) that are traditionally not considered within these studies but that, nevertheless, have a long tradition in the fields of logic, argumentation theory and rhetoric. In the following sections we introduce our case study and we report our results about this aspect.

## 4. Case Study: The Online Magazine

Born from a project of communication renewal of the University of Turin, UnitoNews is an online magazine that aims at presenting and telling our university, its history and its events; at describing and disseminating research groups' activities, national and international projects, collaborations, etc.; and, in general, at reporting exciting news about university life. The contents of the online magazine are created and managed by the Press Office of our university, which daily updates the news, while the entire system has been designed and developed by our research group. The system is developed on a simple LAMP, built on a Linux System with an Apache HTTP Server , MySql database, using PHP programming language. Up this stack there is an instance of a popular CMS, Concrete5 (https://www.concrete5.org/).

The typical target visitors of the magazine are usually people belonging to the university itself (students, teachers and researchers, ICT and administration staff, etc.) but the website is also visited by the "non-academic" world external to the university, such as press agencies and other regular visitors. This type of target can be considered a challenging testbed because, usually, more educated people are more difficult to persuade since education is usually a buffer against biases.

The magazine was launched in November 2015 and, in recent years, it has collected about 6700 users and 11,900 views pages per month from all over the world but particularly from our home country (94%) with 15% of returning visitors with on average 1.50 viewed pages per user and session, and a sessions-to-users ratio of 1.12. From usage reports, 543 users visit the site every day on average with the higher turnout during weekdays, especially during office hours (with some peaks on morning hours, middle of the day and late afternoon). About 52% of users use their desktop/notebook (43% Windows OS, 7% MacOS), but noteworthy is also the usage from mobile devices 48% (58% Android devices, 40% iOS devices).

Users can find and read news by starting from the featured or recent news in the home page, or by navigating through the main categories. However, most of users (46%) visit the site because of the news promotion in the official home page of our university web site (https://www.unito.it/), which

embeds featured news from the magazine UnitoNews (see Figure 1), and (18%) because news and events are shared on the main social networks (e.g., Facebook, Twitter, etc.). Therefore, very often users directly access the news detail page, without going through the home page or other sections. For this reason, the news detail page is an effective point where other news can be promoted to increment users' average time on site.

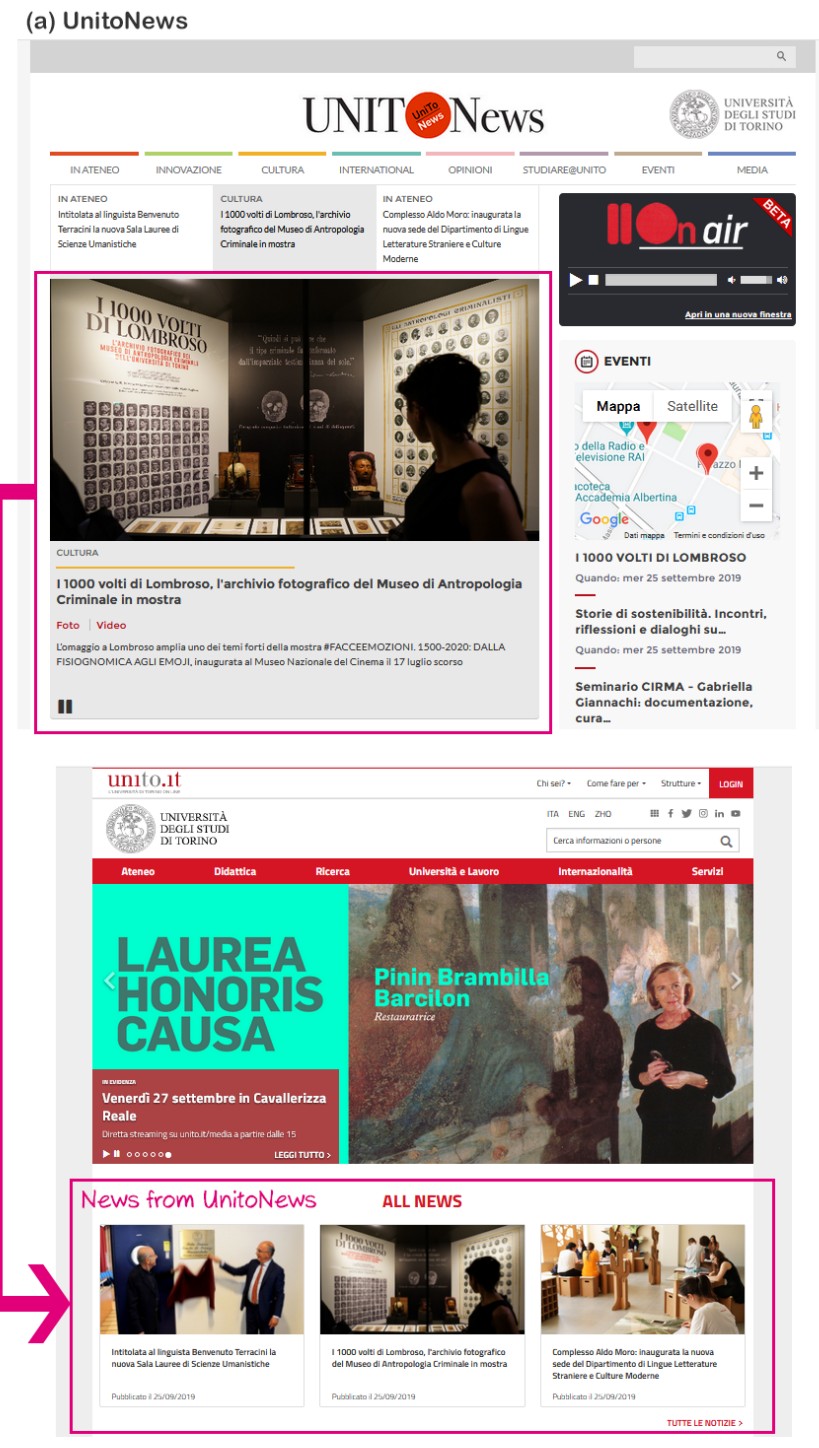

**Figure 1.** The official home page of our university web site (**b**), embeds featured news from the magazine (**a**).

Every news item is enriched by photo or video galleries, inline multimedia contents such as audio players, and social share and likes buttons (Facebook, Twitter, Google Plus, Pinterest). To improve navigation through contents and to promote serendipity effects, on the right sidebar of every news detail page are shown related and recommended contents (see Figure 2), namely the most recent news in the same categories or the most recent news sharing the same descriptive tags.

Notice that even if the online magazine already provides recommended contents, the university management does not want that users must sign in to access the web site, due to internal policies. Hence, the online magazine represents a perfect example of a non-personalized recommender system.

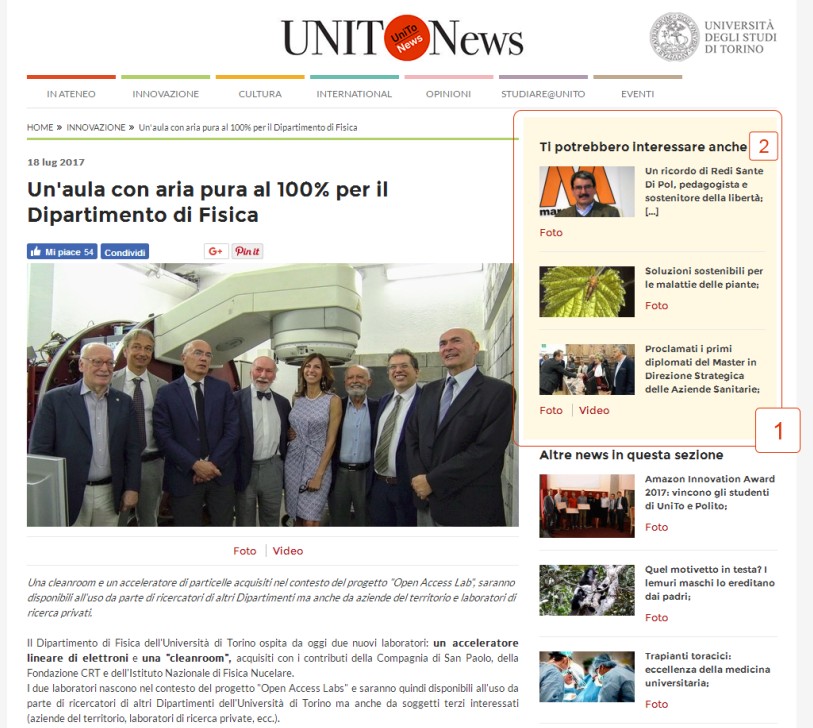

**Figure 2.** News detail page with a set of recommended news (1). Notice the yellow box used to provide visual accent and the persuasive sentence introducing recommendations (2).

## 5. Evaluations Rationale

As already mentioned, in this paper we adopted the conceptual framework on fallacies and persuasive technologies proposed by [14], and conducted two large-scale, real-world empirical evaluations aimed at assessing the effectiveness of different types of persuasive sentences in influencing users' browsing behavior, and, in particular, in promoting user clicks on the recommended content. More specifically, we used persuasive sentences to introduce a short list of recommended news appearing in the detail page of news items, on the right sidebar described in Section 4 (see Figure 2). Notice that we chose to display recommendations in the detail page because it represents the main access point to the magazine contents, as explained in Section 4 and this should therefore guarantee that recommended news is highly visible and easily accessible to users.

It is also important to stress that our evaluations focus only on how news is presented and explained to the user and not on the recommendation itself, whose content does not vary in the different experimental conditions, in order not to influence the results. More specifically, each news item consists of a picture and a short title which are unchanged, irrespective of the experimental condition. In our first evaluation, we used two fallacious sentences and a neutral one, while in our second evaluation we compared two sentences making use of positive and negative framing, respectively.

As we chose a real online magazine as a case study for our evaluation, we were faced with specific opportunities and constraints. On the one hand, we could follow the "evaluation in the wild" approach, according to which empirical studies are carried out in situ and participants are free to use the evaluated application without constraints and for their own situated purposes, while their activities are logged unobtrusively [58]. In both evaluations, participants were regular visitors of the magazine, from whom we did not collect any personal information. Notice that recruiting normal website visitors as participants in our evaluations can be considered equivalent to using an availability sampling strategy. The availability sampling is a sampling of convenience, based on subjects available to the researcher, often used when the population source is not completely defined. Even though random sampling is the best way of having a representative sample, these strategies require a great deal of time and money. Therefore, much research in psychology is based on samples obtained through non-random selection [59]. To this respect, we must also take into account the fact that, as previously pointed out, visitors of the magazine belong for the most part to the university itself (students, teaching and administrative staff), and can therefore be considered to be part of a relatively highly educated niche, which can show peculiar reactions (usually, showing more resistance) to persuasion attempts.

On the other hand, as previously mentioned, the university management decided to avoid user registration and news item rating due to internal policies. Thus, we could only collect user feedback related to their browsing behavior, in the form of clicks, and we were not able to track the long-term behavior of single users, since we could not ask users to log in, but we could only send cookies, which are a partial and incomplete method to model a single user behavior [60], to identify users and keep track of the conditions they explored.

Finally, we want to underline that we only considered the interactions that occurred on the desktop version of the web sites, since the recommended news is hidden at the bottom of the page, below the fold, when it are displayed on the mobile version.

### 5.1. Evaluation: Fallacies

This evaluation aims at studying the persuasive effectiveness of recommendation explanations that leverage popularity. As discussed in Section 2.3, not only do many commercial recommender systems and marketplaces suggest popular items, but also scientific research showed that information on item popularity has an impact on users' choices, basically strengthening their preferences for already popular items (see, e.g., [16–19]). In [14], however, the authors carried out a study in the context of an online bookshop, where they compared different types of affordances aimed at guiding users' behavior, and found that highlighting popular books or annotating them as bestsellers (a form of social navigation) was completely ineffective. Although extremely small-scale and therefore not generalizable in itself, this study somehow anticipated the findings of a larger-scale study carried out on Twitter data [15], and has the merit of raising questions about the usage of persuasion strategies based on the mere popularity, as well as about the factors that might mediate their effectiveness, such as the target or the promoted content. Hence, in this evaluation we focused on the *Argumentum Ad Populum* fallacy as it is normally described in the literature and on a slightly modified variant that we called *Group-Ad Populum*. Surmising that standard popularity-based strategies might not work well in domains where choices strongly involve personal preferences, in our *Group-Ad Populum* we took inspiration from recommendation strategies based on item–item associations by making no appeal to a generic majority, but to the majority of a particular group, i.e., the users who also read the current news item. Thus, we compared the following three persuasive sentences: (1) *You could also be interested in...* (neutral condition); (2) *Similar-to-You Users who read this news also read...* (*Group-Ad Populum* condition); (3) *Most Popular news* (*Argumentum Ad Populum* condition).

Moreover, we wanted to study the effectiveness of the visual accent technique. To this aim, we compared two ways of presenting recommended news, one with no visual enhancements, and one where recommended news was displayed in a yellow box to make it more prominent (see Figure 2).

We hypothesized that:

**Hypothesis 1** (H1)**.** *Users would click on recommended news annotated with the Argumentum Ad Populum sentence more often than on news annotated with the neutral sentence.*

**Hypothesis 2** (H2)**.** *Users would click on recommended news annotated with the Group-Ad Populum sentence more often than on news annotated with the neutral sentence.*

**Hypothesis 3** (H3)**.** *Users would click on recommended news annotated with the Group-Ad Populum sentence more often than on news annotated with the Argumentum Ad Populum sentence.*

**Hypothesis 4** (H4)**.** *Users would click on recommended news with visual accent more often than on recommended news with no visual accent.*

**Design.** $3 \times 2$ within-subjects factorial design, where the independent variables are "persuasive sentence" (three levels: *neutral*, *Argumentum Ad Populum*, *Group-Ad Populum*) and "visual accent" (two levels: *accent* and *no accent*), see Figure 3. The dependent variable is the number of clicks. The recommendations were the same in all the different conditions so that they have the same influence on each of the treatment conditions, and they were updated daily.

**Figure 3.** Experimental design.

**Subjects.** 16,134 user sessions were logged by means of cookies. A power analysis was carried out to determine the minimum sample size to have statistical power above 0.8, following the procedure described in https://www.theanalysisfactor.com/5-steps-for-calculating-sample-size/. Based on hypotheses H2 and H4, the target sample size we obtained ranged between 37 and 1541, far less than our actual sample size. Participants were regular visitors of the magazine, whom we did not collect any personal information about. Participants were initially assigned to a random condition, which was updated after a week. Considering a single user session, each participant was exposed to a single persuasive sentence per week.

**Apparatus and materials.** The magazine was used as a case study (a statement in its footer warns users that the web site may host experiments related to computer science research). Three recommended news items were displayed in the detail page of each article, in the right sidebar (see Figure 2(1)). Persuasive sentences were used as a caption to introduce recommendations (see Figure 2(2)).

In the visual accent condition, the recommended news was highlighted through a yellow box. In our main evaluation phase, recommended news was positioned in the top part of the sidebar, while non-recommended content, introduced by the heading "Other news in this category" was displayed below them.

**Procedure.** Recommended news was selected according to a popularity score *pop* computed as follows:

$$pop = 0.3 * \left( \frac{count(views_n)}{\sum_{i=1}^{N} count(views_n)} \right) + 0.7 * \left( \frac{\frac{count(fbshare_n)}{days_n}}{\sum_{i=1}^{N} \frac{count(fbshare_n)}{days_n}} \right)$$

In our equation, $N$ is the set of news aged at least one day, $count(views_n)$ and $count(fbshare_n)$ are the number of views and Facebook shares received by news item $n \in N$, respectively, and $days_n$ is the age of news item $n \in N$, calculated in days. Aiming at avoiding very popular news that may be promoted elsewhere in the website (e.g., the home page slider), and on the home page of the official university web site and thus users are probably already aware of, the three news with the lowest *pop* scores were chosen for recommendation. Notice that there is no relationship between the content of the persuasive sentences used to introduce recommendations, e.g., "Most Popular news", and the actual popularity of news themselves. Recommendations, which were the same in all the pages of the site regardless of the content of the news, were updated daily to guarantee that: (1) they were fixed during each browsing session and during the day, so that we did not have to deal with possible influences due to news variability; (2) they could follow updates in news popularity.

The magazine was programmed so that we could randomly assign user sessions to experimental conditions, and thus expose users to different persuasive sentences and visual annotations (accent/no accent) in a counterbalanced way. According to the "evaluation in the wild" approach, participants were free to interact with the web site, and interaction logs were collected unobtrusively. Logs contained information on the experimental condition, on the news which was clicked, and whether it was recommended or not.

The evaluation was carried out in winter 2018/2019, specifically from December 4th through February 4th.

**Results.** Recommended news received 4949 clicks, for a total of 74 recommended news. The average number of clicks per news was 20.18 (SD = 22.42) in the *neutral* condition, 19.70 (SD = 23.39) in the *Group-Ad Populum* condition, and 20.48 (SD = 22.58) in the *Argumentum Ad Populum* condition, respectively. Looking at these aggregated data, we found no significant differences between these three different conditions according to the results of Wilcoxon signed-rank test: neutral vs. *Group-Ad Populum* ($Z = -1.244$, $p = 0.213$), neutral vs. *Argumentum Ad Populum* ($Z = -0.327$, $p = 0.744$), and *Group-Ad Populum* vs. *Argumentum Ad Populum* ($Z = -0.495$, $p = 0.620$).

To assess H1, H2, and H3 we separately evaluated the news with and without visual accent, since we wanted to isolate the possible influence of visual accent, and to concentrate on the effect of the persuasive sentences instead.

32 recommended news have been presented without visual accent and received 1558 clicks. The average number of clicks per news was 12.10 (SD = 18.17) in the *neutral* condition, 12.76 (SD = 17.59) in the *Group-Ad Populum* condition, and 13.15 (SD = 17.77) in the *Argumentum Ad Populum* condition, respectively. Again we found no significant differences between these three different conditions according to the results of Wilcoxon signed-rank test: neutral vs. *Group-Ad Populum* ($Z = -0.582$, $p = 0.560$), neutral vs. *Argumentum Ad Populum* ($Z = -0.10$, $p = 0.992$), and *Group-Ad Populum* vs. *Argumentum Ad Populum* ($Z = -0.196$, $p = 0.845$).

The recommended news presented with visual accent were 42 and the total number of clicks on them 3391. The average number of clicks per news was 28.27 (SD = 23.54) in the *neutral* condition, 26.64 (SD = 26.46) in the *Group-Ad Populum* condition, and 27.82 (SD = 24.63) in the *Argumentum Ad Populum* condition, respectively. We found no significant differences between neutral vs. *Argumentum Ad Populum* ($Z = -0.699$, $p = 0.485$), and *Group-Ad Populum* vs. *Argumentum Ad Populum* ($Z = -0.890$, $p = 0.374$). However, we found a significant difference in the neutral vs. *Group-Ad Populum* ($Z = -2.127$, $p = 0.033$) condition. Thus, regarding our hypotheses, we observe that H1 was not confirmed both in the with and without visual accent condition, since we found no differences between the *neutral*

and *the Argumentum Ad Populum* conditions, as confirmed by Wilcoxon signed-rank test (Z = −0.506, p = 0.613 in the former and Z = −0.699, p = 0.485 in the latter condition).

Similarly, our results did not allow us to confirm H2. Indeed, when the visual accent was not present we did not find significant differences between the *Group-Ad Populum* and the *neutral* condition, as reported by Wilcoxon signed-rank test (Z = −0.582, p = 0.560). In the visual accent condition, the difference between the *Group-Ad Populum* and *neutral* condition was significant (Z = −2.127, p = 0.033), but due to a larger number of clicks in the neutral condition.

Regarding the hypothesis H3, we found no differences between the *Group-Ad Populum* and the *Argumentum Ad Populum* conditions both in the *with* and in the *without visual accent* condition, as confirmed by Wilcoxon signed-rank test (Z = −0.890, p = 0.374 in the with and Z = −0.196, p = 0.845 in the without condition). Thus, our intuition that the *Group-ad Populum* fallacy, since it refers to a particular group rather than to a generic "majority" of people, may receive more attention from the users than the *Argumentum Ad Populum* fallacy was not empirically confirmed. A possible explanation could be that the *Group-Ad Populum* fallacy is so often used in e-commerce platforms and entertainment sites that it has perhaps lost its persuasive effectiveness, as also suggested by the results of H2.

Finally, regarding H4, Wilcoxon signed-rank tests highlight the existence of significant differences in the *neutral* (Z = −3.221, p = 0.001), *Group-Ad Populum* (Z = −2.381, p = 0.017), and *Argumentum Ad Populum* (Z = −2.663, p = 0.009) conditions with and without visual accent, respectively. Also aggregating all the persuasive sentences and considering just the difference between the **with** and without visual accent conditions produces significant results (Z = −4.719, p = 0.000). Thus, we can conclude that, as expected, the presence of visual accent attracts users' attention and causes an increase in the number of users' clicks when the presented content is the same.

*5.2. Evaluation: Framing*

In our second evaluation, we compared the following two sentences, vehiculating two differently polarized frames: (1) *This week's featured news* (vehiculating, with a positive framing, the fact that the recommended news has something special with respect to the others), and (2) *Featured news only available until [...]* (a sentence vehiculating the reverse of the previous frame, i.e., a negative framing). The negative framing sentence was dynamically completed with next Saturday's date, and was conceived to suggest a mild sense of "loss" in case users were not quick to profit from the recommendations and read the suggested news. In both cases, we used the same visual accent technique we adopted in our previous evaluation (see Figure 4) since our results have confirmed the efficacy of this technique in raising users' interest.

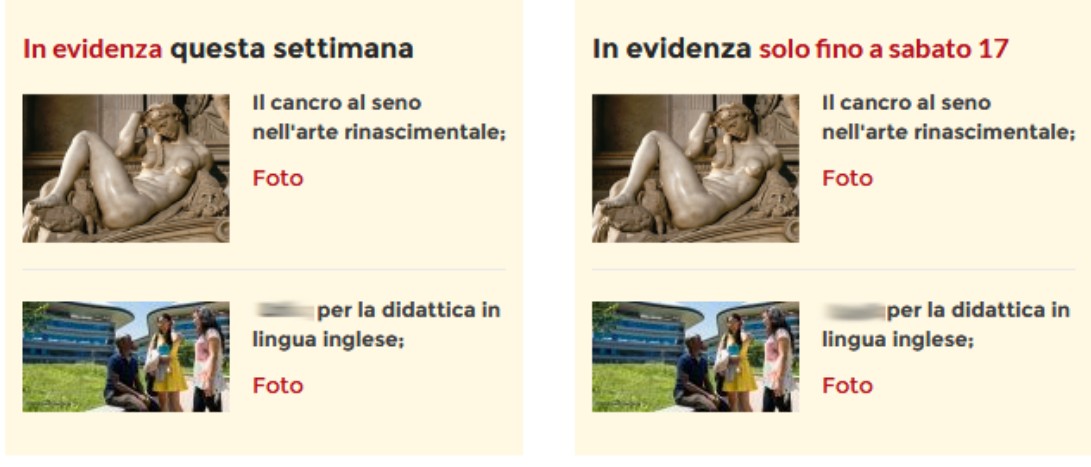

**Figure 4.** Recommended news with positive framing (**left**) and negative framing (**right**).

We hypothesized that:

**Hypothesis 5** (H5). *Users would click on recommended news annotated with the negative framing sentence more often than on news annotated with the positive framing sentence.*

**Design.** Within-subjects design, where the independent variable is "persuasive sentence" (two levels: *positive framing* and *negative framing*). As with our previous evaluation, users participate in both experimental conditions. The dependent variable is the number of clicks. As in our previous evaluation, the recommended news was the same in the two conditions, and it was updated weekly.

**Subjects.** 20.933 user sessions were logged. As with our previous evaluation, a power analysis was carried out in order to determine the minimum sample size to have statistical power above 0.8. Based on hypothesis H5, the target sample size we obtained was 2061, far less than our actual sample size.

**Apparatus and materials.** The magazine was used as a case study. Two recommended news items were displayed in the detail page of each article, in the right sidebar (see Figure 4). Persuasive, framing-based sentences were used as a caption to introduce recommendations. As with our previous evaluation, recommendations were the same in all the pages, regardless of the content of the news.

**Procedure.** Each Saturday night, "fresh" recommended news was randomly selected among the news published in the previous week. Cookies were used to identify users and keep track of the condition they were assigned to counterbalance the exposure to the conditions.

Also, in this case, all the participants received the same recommendations. Participants were free to interact with the web site and interaction logs were collected unobtrusively, in accordance with the "evaluation in the wild" approach. Logs contained information on the experimental condition, on the news which was clicked, and whether it was recommended or not.

The evaluation was carried out in spring 2019, specifically from 5 April through 5 June .

**Results.** Recommended news received 2853 clicks, for a total of 97 recommended news, which were presented both in the positive and negative framing conditions.

The recommended news presented in the negative framing condition received a total number of clicks of 1504, while in the positive framing condition we found that the same number of recommended news only received 1349 clicks.

The average number of clicks per news was 15.50 (SD = 29.76) in the *negative framing* condition and 13.90 (SD = 27.13) in the *positive framing* condition, respectively.

Regarding our hypothesis (H5), Wilcoxon signed-rank test confirmed that users would click on recommended news annotated with the *negative framing* sentence more often ($Z = -2.351$, $p = 0.019$) than on news annotated with the *positive framing* sentence, and so we rejected the null hypothesis. Thus, empirical results confirm that the use of sentences implicitly suggesting a sense of loss are more effective than the more neutral ones.

### 5.2.1. Survey

At the end of our second evaluation, we invited the magazine users to participate in a short survey. Our main goal was to have an empirical confirmation of the audience composition of the magazine and to understand if they had actually noticed the recommended news. The survey was advertised through a banner in the magazine homepage and a direct email message to the staff of our university, and was available for a few days.

Our results show that 461 people completed the survey. Almost all of them were adults (91% >18 years) belonging either to the administrative/technical staff (41.5%) or to the teaching/research staff (43.2%) of our university.

Only 37.1% of the survey respondents can affirm they have received recommendations (see Figure 5), either in the "This week's featured news" form, or in the "Featured news only available until [...]" form (even if they were actually provided). According to this datum, it seems that this kind of recommendation is scarcely perceived as intrusive since it is explicitly designed to follow the so-called peripheral route of the well-known Elaboration Likelihood Model (ELM) elaborated by [61]. The ELM theory, in fact, describes two different information processing paths: one, the peripheral route, where the processing is based on scarce attention and on the focus on surface elements (and as such more akin to trigger fast and automatic cognitive mechanisms that are not subject to any form of deliberative control) and another one, the central route, through which the information is processed in a more deliberative, controlled and logical way. Fallacious arguments and persuasive techniques, such as the ones considered in these studies, exploit the peripheral route and, as such, are assumed to be processed automatically thus avoiding the cognitive control of the central route of information elaboration. This is precisely the reason both fallacious arguments and framing-based techniques are effective from a persuasive point of view and this aspect seems confirmed by our findings.

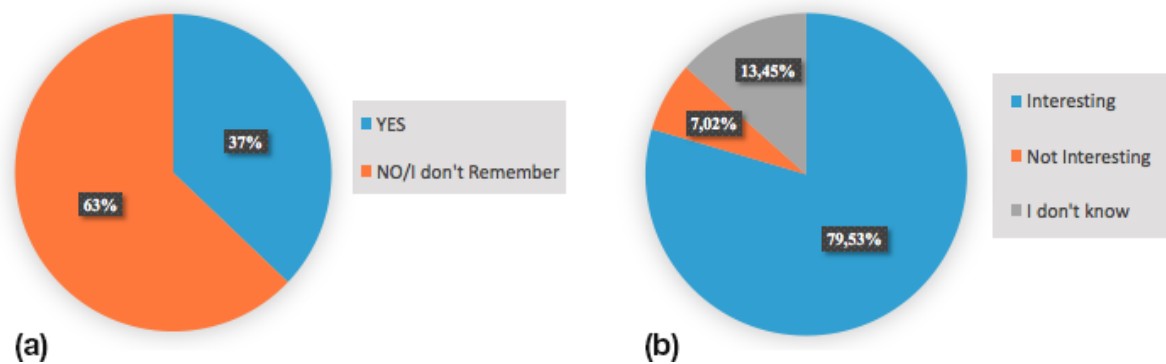

**Figure 5.** Users remembering at least one type of recommendations (**a**), and user perceptions of their interestingness (**b**).

In addition, respondents were asked to assess recommendation quality: 79.53% of participants who had noticed at least one type of recommendations judged the suggested contents interesting (see Figure 5). We consider this result encouraging, in that it suggests that non-personalized recommendations can be successfully included in websites that cannot afford personalized recommendations, due to their content access policies or other constraints. However, in interpreting this specific result, we must also take into account the fact that response biases, such as the social desirability or acquiescence biases [62], might have influenced participants' answers, especially because most participants belong to the same organization that promoted the survey.

## 6. Discussion

Carrying out evaluations "in the wild" implies specific opportunities as well as criticalities. As pointed out in [58], research in the wild shows "how people come to understand and appropriate technologies on their own terms and for their own situated purposes", which is valuable with regards to our general goal of understanding whether the use of persuasive explanations can promote user clicks on the recommended content, in a real usage scenario. On the other hand, it is certainly more difficult to isolate specific effects in comparison with a standard controlled experiment, and all our results should be interpreted bearing in mind that they were collected "in the wild".

In the following, we will discuss some aspects that are relevant for the design of our evaluations, and explain how we have tried to minimize possible confounding factors.

**Study design.** In both our studies, we used a within-subjects design. Participants were initially assigned to a random condition, which was updated after a week. Thus, each participant was exposed to a single persuasive sentence per week, along a single user session. Cookies were used to identify

users and keep track of the condition(s) they were assigned to. As already acknowledged, the use of cookies to identify and track participants presents some limitations [60], since it only allows gain of a partial and limited understanding of their online behavior. However, they were the only means, agreed with the university management, that we could use in the present large-scale study. A specific problem connected with the use of cookies regards the fact that the same user might engage in multiple browsing sessions during the evaluation period, thus interfering with our management of condition assignment. We are confident, however, that there is only a remote possibility for such a case to actually happen: in fact, not only have we focused our evaluation only on the interactions that occurred on the desktop version (thus preventing the case that the same user might engage in parallel browsing sessions from mobile and desktop devices), but the specific features of UnitoNews readership also make the chance that the same participant might use different browsers very unlikely. As we discussed in Sections 4 and 5.2.1, in fact, most users are members of either the administrative/technical or the teaching/research staff, who access the magazine during office hours using their workstation, where they usually find pre-installed software (e.g., a single default browser) and only have limited personalization possibilities.

**Participants.** Participants in our evaluations are actual readers of UnitoNews, for the most part members of the staff of our university (see Sections 4 and 5.2.1) and therefore a relatively highly educated niche. While the specific features of UnitoNews readership may raise questions on the generalizability of our results, we maintain instead that they can contribute to create some sort of highly challenging, "worst case" scenario, where participants are expected to show more resistance to persuasion attempts.

**Possible confounding factors.** Participants in our study may have decided to access UnitoNews out of different reasons, among which are certainly authority (e.g., when a piece of news is featured in the official home page of our university web site), social proof (e.g., when a piece of news is shared on social networks), and trust (in both cases). With a bounce rate over 80%, however, most users abandon the website after accessing a single page. In our evaluation, we concentrated on the behavior of users who continue browsing beyond the landing page, taking advantage of the website navigation features and, in particular, of the list of recommended news. While arguments of authority, trust and social proof can have played a role in bringing users to the magazine, there are no specific reasons to assume they have also influenced their following decision to explore (or not) recommended news (which were not promoted in social networks). Authority and trust may still have influenced the readers' behavior at a general level, but this is an intrinsic feature for a credible and respected information source. On the other hand, since participants' behavior was observed unobtrusively, it is very unlikely that participants may have changed their behavior out of social desirability reasons, so as to appear more "compliant".

Our results also call for some discussion. Our finding that negative framing has a positive effect on the number of user clicks confirms our expectations and has a strong theoretical grounding (see, e.g., [13,53]). Similarly, the fact that visual accent attracts users' attention and causes an increase in the number of clicks is compliant with research on visual biases [63], as well as with standard interaction design best practices. In contrast, our finding that the adoption of the *Argumentum Ad Populum* and *Group-Ad Populum* fallacies seems unable to influence user choices is quite surprising, even if it confirms the results of some previous works (see, e.g., [14,15]). In fact, not only the scientific literature on fallacies [46], as well as empirical studies (see, e.g., [16–19]) have confirmed the persuasive power of popularity arguments, but also commercial recommender systems make heavy use of the "appeal to the majority" for the recommendation of movies, news, and other products. In the following, we will discuss some of the factors that may explain this result.

**Participants.** As already mentioned, we can consider the readership of UnitoNews to be a relatively highly educated niche, less prone to succumb to persuasion attempts. This might be especially true when arguments are used that refer to "popularity", which participants may prefer to disregard in accordance with their self-image of well-read, independent thinkers.

**Content.** The effectiveness of popularity arguments might depend on the promoted content. News, books [14] and updates from social connections [15] might be domains where personal preferences play a prevalent role. In addition, it might be easy for users to assess whether a certain piece of news is worth reading, for example based on its topic and title, and the cost of a wrong decision is limited: thus, users might find it unnecessary to turn to the "wisdom of the crowd" to make a choice. This is not the case in other domains: for example, when buying consumer electronics, users might find it difficult to assess all the relevant features and compare different products.

**Task.** In controlled experiments, participants are usually asked to carry out a task where they have to make some decision that explicitly involves the "popular" items, such as in Zhu and Huberman [18], where participants were asked to choose their favorite options from pairs of items, with and without popularity-related information. In contrast, in evaluations in the wild, participants are free to carry out their own tasks: it might be the case that in online magazines such as UnitoNews, these do not even involve the possibility to explore other news, for example due to reading habits or time constraints. The high bounce rate (over 80%) of UnitoNews actually seems to suggest a pattern where users only read a single page per visit. In this context, successful persuasive explanations should not only make an option more attractive than others, but also suggest a goal that users would not normally consider.

**Popularity blindness.** Banner blindness describes the tendency to ignore page elements that are perceived as ads (https://www.nngroup.com/articles/banner-blindness-old-and-new-findings/). Users might have developed a similar selective behavior with regards to popularity-based persuasive contents, consequent to their abuse. In might also be the case that people have learnt to associate persuasion attempts to the use of popularity-based arguments, and are therefore less likely to be persuaded than if they had no prior knowledge [64].

## 7. Conclusions and Future Work

In this paper, we presented the results of two large-scale evaluations, conducted with real users on a real online magazine, aiming at investigating the efficacy/inefficacy of some fallacy-based persuasive techniques in news recommendations.

A relevant result emerging from our analysis is that the adoption, for persuasive purposes, of the standard *Argumentum Ad Populum* fallacy, as well as of the *Group-Ad Populum* fallacy, is not effective, at least in a news-recommendation context. In Section 6 we discussed some of the factors that may contribute to explain this result, which is prone to open a debate on the actual persuasive power of arguments based on an "appeal to the majority", with some relevant related works (see, e.g., [16–19]) confirming, and some others (see, e.g., [14,15]) denying it.

In contrast, coherently with our expectations, negative framing appears to be effective at promoting user clicks on recommended news, in comparison with its positive counterpart, thus suggesting that sentences implicitly evoking a sense of loss are more effective than more neutral ones.

As future work, we aim at improving our understanding of user perceptions of persuasive recommendations. To do so, we are planning to carry out a more fine-grained and controlled experiment in a laboratory setting, where we will be able to avoid confounding factors that usually characterize "real-world" studies, and ask participants to perform browsing tasks while following the *thinking aloud* protocol. Even if a controlled experiment is carried out in an artificial setting, we believe that it can be useful to understand participants motivations and perceptions of the recommended content. More specifically, data collected in the controlled experiment might be coupled with data from our current evaluation in the wild to gain a better understanding of persuasion dynamics.

In addition, we plan to investigate the efficacy of more complex forms of fallacy-based persuasion. Moreover, we want to extend the number of considered fallacy-reducible strategies, e.g., by including the *Argumentum Ad Verecundiam* (also "appeal to the authority", see [14] and Section 2 for details) and the *Argumentum Ad Consequentiam* (see [14] for details) and other configurations of the *Framing effect*.

Finally, another relevant and interdisciplinary line of investigation that deserves to be detailed in the near future concerns the assessment of the typology of uses that are considered ethically acceptable

for this kind of techniques in the area of persuasive technologies. We maintain that the adoption of these techniques should entirely fall within the so-called *Ethical Framework for a Good AI Society* [65]. Therefore, their integration should follow an "ethics by design" approach, in which ethical and social considerations are taken into account at the design phase of any given system. Despite the agreement with such general principles, however, we are aware that a more detailed analysis of such elements is needed and requires deeper investigations.

**Author Contributions:** Conceptualization, C.G., P.G., A.L., C.M. and F.V.; methodology, C.G., P.G., A.L., C.M. and F.V.; software, C.G., P.G., A.L., C.M. and F.V.; validation, C.G., P.G., A.L., C.M. and F.V.; formal analysis, C.G., P.G., A.L., C.M. and F.V.; investigation, C.G., P.G., A.L., C.M. and F.V.; writing–original draft preparation, C.G., P.G., A.L., C.M. and F.V.; writing–review and editing, C.G., P.G., A.L., C.M. and F.V.

**Funding:** This research received no external funding.

**Conflicts of Interest:** The authors declare no conflict of interest.

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
