# Peer review of "When Personalization Is Not an Option: An In-The-Wild Study on Persuasive News Recommendation"

_information, doi:10.3390/info10100300_

Round 1

Reviewer 1 Report

Line 108… In the motivation for the Group Ad Populum, it is stated that recommending a bestseller in a website is a useless approach. Evidence for this is a small-scale empirical evaluation [14]. Results in literature are mixed, so I would encourage a more thorough argumentation / examination of related works.

Line 202-210. Either provide references to the described experiments or describe them in full. Readers cannot accept the statements as facts without some evidence.

Line 318-365. Since you want to take us on board that there are no significant interests, you shoud demonstrate that the test was powerful enough to uncover such differences , e.g. statistical power above 0.8

Discussion section: while a lot of related research is reviewed it is not clear to me what you consider the design implications of your work, and how you explain that your results are not consistent with prior art. Perhaps reflect on what kind of data you would need to collect to provide such an explanation. A good stab at this is in the ‘conclusions’ section. I would prefer a slightly more elaborate and documented argument to be the discussion section. The current discussion section seems to me to be only partially interesting as related work (could follow the introduction). In any case, the information presented is too extensive and of little relevance.

Author Response

We are very grateful to Reviewer 1 for their insightful comments.

Reviewer 2 Report

The paper presents a real-life study into the effect of persuasive sentences and presentation aspects (e.g. highlighting) on online (clicking) behavior.

Comments on writing

In general, the paper is well written. There are some minor textual comments:

References should be checked. For example, the reference 4 and 46 are identical. line 179: "visitors with 1.50 viewed pages per user and session" this is probably on average so I would rephrase as "visitors with on average 1.50 viewed pages per user and session" line 233: "on how news are presented" should be "on how news is presented" line 245: "magazine, whom we did" should be "magazine, from whom we did" line 245: "recruiting normal website visitors" should be ""recruiting regular website visitors" line 425: "Related works" should be "Related work"

Comments on the study validity

As a reviewer, I have some serious problems with the study as there are many different potential confounding factors (hence my low score on scientific soundness).

1. The role of persuasive sentences

The authors have adopted to use persuasive sentences (based on logical fallacies) alongside news items. More specific, the news item was displayed with a statement such as "popular item" alongside a picture and abstract of the news item. In this sense one cannot know for sure that the item was selected based on the persuasive sentence or the picture & abstract of the item.

2. The study sample

As acknowledged by the authors, the majority of the study sample was recruited from the same organisation as the one presenting the news magazine. While such bias could be minimised by taking the position that the study findings cannot be generalised beyond the specifics of the sample, for a study on persuasion this is not so simple. Specifically because arguments of authority (i.e. the organisation that is employing the participant) and trust (as confirmed by the authors as an important mechanism (see line 515)) could have influenced the participant's behavior. 

3. Independence of conditions

Given that is a study in the wild, the authors acknowledge limitations. However, not having control over the experimental conditions is very problematic (and basically invalidates any conclusion). More specific, given the study design it was very well possible that the same user would have been in different experimental conditions (the authors talk about "user sessions" and indeed not "users"). Even stronger, if the same user would have visited the same page multiple times on the same day it could have been the case that they were presented the same highlighted article (since highlighted articles are changed only once per day) with different persuasive sentences. In that case, one would learn more about a user's susceptibility to a persuasive sentence than the persuasive power of the sentence. But given that there seems to be no control over which condition a user is in, even this conclusion cannot be made. If it is indeed the case that there was no experimental control (as acknowledged on line 292) then I fail to see how the authors can correctly come to any conclusion from the study.

4. Referral to the study site

The fact that many study participants (18%) would access the news magazine based on an item shared in their social network does raise the question if an argument of social proof was involved in bringing participants to the respective news item. This would be a serious confounding factor in the study design.

5. Response bias

Finally, there could have been response bias in the survey asking participants how interesting the recommendation were. Of course such bias is always potentially present in any study, but given that the study site is part of the same organisation as the study participants, could increase the potential of response bias.

Comments on operationalisation

Besides the serious limitations due to the above mentioned potential confounding factors, there are some issues related to the operationalisation of some study concepts.

1. Logic fallacies

I am not sure that the Group-Ad Populum and Argumentum Ad Populum condition are very distinct. For the former, the operationalisation "similar to you" was chosen while for the latter the statement "most popular" was chosen. I wonder if both are semantically different (for the study participants). The statement "similar to you" reflects a similarity in choice but it is not per-se reflecting a similarity in person. A stronger statement would have been "people with your interest" or "people with your profile". I understand that it could be a limitation due to the fact that user registration was not possible (and thus no profile to refer to) but it might be an explanation why the authors did not find a significant effect.

2. Positive / negative framing

The operationalisation of positive / negative framing is implemented as "featured" versus "timely" (i.e. a scarcity argument). I am not sure if this does reflect a positive versus negative framing. The negative framing could have been an implementation of the scarcity argument...

General comments

One aspect that is not discussed by the authors is the role of the content itself. On line 577 a study for e.g. movies is mentioned. Could it not be the case that a movie preference is stronger in reflecting a personal preference than a news article? In any case, the authors could discuss the potential influence of content.

The related work section is well written and interesting, but for some of the mentioned theoretical background it is not clear why this is part of this section (e.g. the footprint approach) since it is not clear how it relates to the actual study reported in the article.

The authors state that "Our results empirically confirm the commonsense intuition that visual accent successfully improves the number of clicks on recommended news.". This is not only a commonsense (and as quoted by the authors part of a design guideline that is 18 years old) intuition but standard practice. Hence, my low score on novelty.

Author Response

We are very grateful to Reviewer 2 for their insightful comments.

Round 2

Reviewer 1 Report

I am happy with the improvements made.

Reviewer 2 Report

I thank the authors for adequately addressing the comments and suggestions made during my review.